# IMU-Based Monitoring for Assistive Diagnosis and Management of IoHT: A Review

**DOI:** 10.3390/healthcare10071210

**Published:** 2022-06-28

**Authors:** Fan Bo, Mustafa Yerebakan, Yanning Dai, Weibing Wang, Jia Li, Boyi Hu, Shuo Gao

**Affiliations:** 1Smart Sensing Research and Development Center, Institute of Microelectronics, Chinese Academy of Sciences, Beijing 100029, China; bofan@ime.ac.cn (F.B.); wangweibing@ime.ac.cn (W.W.); 2School of Microelectronics, University of Chinese Academy of Sciences, Beijing 100049, China; 3Department of Industrial and Systems Engineering, University of Florida, Gainesville, FL 32611, USA; mu.yerebakan@ufl.edu; 4School of Instrumentation and Optoelectronic Engineering, Beihang University, Beijing 100191, China; 16171056@buaa.edu.cn; 5Beijing Advanced Innovation Center for Big Data-Based Precision Medicine, Beihang University, Beijing 100191, China

**Keywords:** Internet of Health Things (IoHT), IMU, machine learning, motion monitoring, disease diagnosis

## Abstract

With the rapid development of Internet of Things (IoT) technologies, traditional disease diagnoses carried out in medical institutions can now be performed remotely at home or even ambient environments, yielding the concept of the Internet of Health Things (IoHT). Among the diverse IoHT applications, inertial measurement unit (IMU)-based systems play a significant role in the detection of diseases in many fields, such as neurological, musculoskeletal, and mental. However, traditional numerical interpretation methods have proven to be challenging to provide satisfying detection accuracies owing to the low quality of raw data, especially under strong electromagnetic interference (EMI). To address this issue, in recent years, machine learning (ML)-based techniques have been proposed to smartly map IMU-captured data on disease detection and progress. After a decade of development, the combination of IMUs and ML algorithms for assistive disease diagnosis has become a hot topic, with an increasing number of studies reported yearly. A systematic search was conducted in four databases covering the aforementioned topic for articles published in the past six years. Eighty-one articles were included and discussed concerning two aspects: different ML techniques and application scenarios. This review yielded the conclusion that, with the help of ML technology, IMUs can serve as a crucial element in disease diagnosis, severity assessment, characteristic estimation, and monitoring during the rehabilitation process. Furthermore, it summarizes the state-of-the-art, analyzes challenges, and provides foreseeable future trends for developing IMU-ML systems for IoHT.

## 1. Introduction

Patients suffering from neurological and musculoskeletal diseases have different levels of mobility. The ones with higher levels of mobility impairments suffer from physical pain stemming from their condition and a lower quality of life, both debilitating to their mental health. To help them regain their independence, plenty of medical institutions provide motion monitoring and evaluation-based rehabilitation services, such as disease diagnosis and exercise training [1,2].

Conventionally, camera-based three-dimensional (3D) motion capture systems [3] and pressure sensor-enabled electronic walkways [4] have been used to monitor body motions for diverse medical evaluation purposes. The former track the positions of markers placed on body segments and form a segmented 3D pose; in contrast, the latter quantify plantar stress distributions for a gait analysis. Both setups are expensive and restricted to laboratory settings; hence, they are mainly employed in medical or research institutions. Due to multiple reasons (limited healthcare resources, challenges of commute for some physical disabled patients, etc.), treatment or rehabilitation in medical institutions might not be viable for every patient. Especially for patients that require longer rehabilitation periods, this is a significant issue. In order to make medical monitoring and rehabilitation more accessible to patients, new methods based on the Internet of Things (IoT) are currently being explored [5,6,7].

The IoT’s application in this domain mainly aims for real-time data collection and providing physicians the necessary information to track their patients reliably. Internet of Health Things (IoHT) systems, such as normal IoT systems, are made up of data acquisition, the user interface, and cloud component [8], data that are collected from wearable sensors and processed by smartphones, tablets, and even smartwatches. This data is then transmitted to a central server, from which the physician can access the patient’s information and make judgments [9]. Islam et al. [10] predicted many potential areas for IoT-based healthcare, such as frameworks for elderly patients that provide more continuous assistance, called ambient assisted living, community-wide health monitoring, and applications such as monitoring a wide variety of vitals such as blood sugar, body temperature, and oxygen saturation. In addition to monitoring vitals, several IoHT systems were proposed for rehabilitation and disease monitoring. Dobkin et al. [11] proposed an in-home Rehabilitation Internet of Things framework that focuses on the improvement of motor functions. Frameworks such as Dobkin’s rely on a wide array of metrics, depending on the type of illness that is being monitored or the circumstance of rehabilitation. For example, metrics such as activity patterns and gait have been proposed to be utilized for monitoring illnesses such as Parkinson’s disease and osteoarthritis [12]. There is also the possibility of monitoring multiple illnesses at the same time. AbdulGhaffar et al.’s proposed system monitors three illnesses: hypertension, glaucoma, and chronic obstructive pulmonary disease (COPD), by monitoring the patient’s blood and intraocular pressures and the blood saturation levels [13].

IoHT systems rely on a wide variety of wearable sensors to collect patient data. Broadly utilized wearable motion-sensing technologies include electromyography (EMG), optical fiber sensors (OFS), Radio Frequency Identifications (RFID), and inertial measurement unit (IMU)-based systems. Among them, EMG reflects users’ motions by capturing the physiological metrics of muscle activity [14,15]. However, the sensitive nature of the sensors makes them prone to noisy data collection, and poor wearability limits their usage in daily life. OFS measures the bending angle by calculating the light intensity attenuation of the optical fiber during a motion [16]. RFID offers body segment inclination using dual-polarized antennas [17]. Both OFS and RFID technologies suffer from low sensitivity and can only offer limited motion information [18]. Compared to the aforementioned technologies, IMUs offer direct motion measurements such as acceleration, angular rate, and magnetic field, which is a convenient and cost-efficient way for continuous human motion detection. Hence, IMU-based technologies have the potential to be widely utilized in future IoHT-based telerehabilitation applications.

Traditional methods to interpret IMU data are based on numerical calculations. However, IMUs suffer from considerable measurement noise and drifting issues, resulting in severe motion misregistration [19,20,21]. To address this issue, researchers have attempted to use machine learning (ML) approaches to find relationships between the messy IMU data and diverse neurological and musculoskeletal diseases. In [22], support vector machine methodology is used to classify different walking conditions for hemiparetic subjects for the gait analysis. Lee et al. [23] utilized the expectation-maximization clustering algorithm to scale the clinical scores of dyskinesia. Inspired by emerging deep learning approaches, [24] proposes a CNN-LSTM model to detect the freezing of gait of Parkinson’s disease patients. Furthermore, a linear regression model is used in [25] to provide accurate joint loading values for hip osteoarthritis patients. The selection of articles above represents only a small fraction of the current prolific output of studies that develop ML algorithms to interpret IMU data for disease analysis and rehabilitation. Researchers and medical staff have increasingly been exploring how artificial intelligence technologies can help in assistive disease diagnosis. The basic generalized workflow of these approaches is illustrated in Figure 1.

However, there is still an enigma when facing a specific monitoring and assessment issue with an ocean of potential data processing methods and not knowing which is the most efficient and suited. The same confusion arises when choosing from different sensor attachment locations, IMU sensor types, feature selection strategies, and evaluation metrics. In order to illustrate the recent developments and outcomes in this field, introduce professionals to this line of research, and inspire more colleagues towards conducting research in this direction, we decided to compose this review of studies that have combined IMU and ML algorithms for body motion monitoring for diseases analysis. In particular, we aim to clarify the following issues in our review:(1)To inventory and classify various ML methods to process IMU data and locate the widely used and state-of-the-art methods regarding different application scenarios and tasks.(2)To inventory the target disorders that can benefit from IMU-ML systems based on movement-related medical conditions that regard specialized areas of rehabilitation.(3)To gauge the implementation details to build IMU-ML systems for assistive diagnosis and management, such as feature selection strategy, sensor attachment selection, and evaluation methods.

The remainder of this paper is structured as follows. Section 2 proposes the methodology of article selection and taxonomy of the selected works. Section 3 summarizes the traditional and mainstream ML methods for monitoring body movements. Section 4 presents the results of our review from the view of different diseases. The findings in this section are organized by the types of disorders. Each type of disorder is complemented with a table that provides detailed parameters obtained per article. In Section 5, we identify the main challenges and the most promising future directions. Table A1 lists all the acronyms.

## 2. Methods and Taxonomy of Existing Approaches

Four databases, including PubMed, Web of Science, IEEE, and ACM, were selected. The search string was created as (inertial sensors OR inertial measurement unit OR IMU OR accelerometer OR gyroscope) AND (machine learning OR deep learning) AND (monitor OR assessment OR motion OR movement OR locomotion) AND (healthcare OR rehabilitation OR physical therapy). The above terms were used to search the titles and abstracts of the articles. The search was conducted on 13th June 2021, and we excluded articles published before 2016.

After retrieving the articles, first, the duplicates were removed. A total of 610 articles were screened in three stages. In the first stage, the titles and, in the second stage, the abstracts of the articles were reviewed. A total of 140 articles were selected for full-text review. Since the focus of the article is using wearable IMUs and ML methods to monitor patients with certain diseases in the process of healthcare, non-directly relevant topics such as the identification of activities of daily living, functional mobility tests, and recognition of rehabilitation exercise were excluded from this review. Moreover, articles where inertial sensors were not used as a dominant part of the analysis were also excluded from this review. Finally, the review resulted in the identification of 81 target studies.

### Taxonomy Structure

To better understand how IMUs and ML methods are used in healthcare applications, we provide a taxonomy based on the selected articles to connect the fields of sensing technology, signal processing, and physical medicine. Similar to the categorization of the disorders in [26,27], the articles included in this review can be classified into four main categories based on movement-related medical conditions in regard to specialized areas of rehabilitation: (1) neurological disorders, (2) musculoskeletal disorders, (3) mental health disorders, and (4) other general disorders, as illustrated by the taxonomy shown in Figure 2.

(1)Neurological disorders, which are the most common disorders that require rehabilitation, arise from people with diseases, injuries, or dysfunctions of the nervous system. Along this line, five kinds of conditions can benefit from neurological rehab: (a) degenerative disorders, such as Parkinson’s disease, multiple sclerosis, and Huntington’s disease, among which, Parkinson’s disease is the most common disorder in all the selected articles; (b) vascular disorder, which is mainly a stroke; (c) neurodevelopmental disorders, which are mainly cerebral palsy; (d) trauma, such as traumatic brain injury, spinal cord injury, and brachial plexus injury; and (e) functional disorders, such as seizure and vestibular system disorders. Additionally, there are other disorders and symptoms presented, and we categorized them as other neurological disorders.(2)Musculoskeletal disorders, including impairments or disabilities due to disease, disorders, or injuries to the muscles, tendons, ligaments, or bones. Three representative conditions can benefit from musculoskeletal rehab: (a) arthritis, which is mainly osteoarthritis, (b) back pain, and (c) Total Joint Replacement (TJR), such as total hip arthroplasty and total knee replacement.(3)Mental health disorders affect a person’s behaviors, feelings, and overall wellbeing, affecting many aspects of their daily lives. This mainly includes depression; however, illnesses such as bipolar disorder and schizophrenia are also represented in the studies included here.(4)Others consist of cardiac disorders, pulmonary disorders, and general rehabilitation, focusing on body parts such as the joints, upper limbs, and lower limbs.

Despite the difference in rehabilitation metrics and targets, IMU- and ML-based methods have been demonstrated to allow for better monitoring and assessment during the whole rehabilitation program. In the following sections, we will discuss these methods in detail.

## 3. IMUs for Monitoring Body Motion

Traditionally, the assessment of various motion impairment diseases has mainly been done by subjective methods, such as questionnaires and visual observation-based evaluations [28]. The first relies on patients’ memories and perceptions, and the latter is based on clinicians’ experiences, both susceptible to biases and inaccuracies. In addition, tasks such as motor activities recognition, the discrimination of symptoms, and events prediction require the supervision of expert clinicians, bringing inconvenience and high costs to patients. IMU-based movement monitoring can offer a solution to both problems by offering more objective measurements at a much cheaper cost to patients. Inspired by the emerging data-driving solutions that can learn a general motion model from a large amount of inertial data without hand-engineering effort, researchers are aiming to establish a trustworthy assistant for diagnosis and assessment in the setting of rehabilitation. The body parts mentioned in the reviewed articles for sensor placement are labeled as demonstrated in Figure 3. Below, two kinds of methods for analyzing IMU data are explained.

### 3.1. Numerical Methods

Numerical methods utilize the acceleration and angular velocity measured from an IMU to calculate the orientation, velocity, and position of body parts [29,30,31]. The basic diagram of the motion tracking system is depicted in Figure 4. Among the parameters, the position and orientation of an IMU are of special importance for body segments and joint analysis, but precisely calculating them is a challenge, due to the offset fluctuation and measurement noise-induced integration drift [32]. To obtain the desired accuracy level, the following methods are normally utilized, including bias compensation, distortion rejection, alignment, filtering (e.g., complementary filter (CF) [33] and Kalman filter (KF) [34]), and zero-velocity update (ZUPT) [35].

After obtaining the position and angle information, previously established movement models have been widely used to estimate high-level parameters for comparing between patients and healthy controls [36]. Linear, biomechanical, context-based, and some adaptive models are the major methods for extracting motion signatures and analyzing motion patterns to provide optimized estimations for a given set of measurements. However, such prior knowledge-based modeling processes inevitably introduce inaccuracies, since the manually extracted features are challenging to fully reflect the diversity of each patient’s motion pattern [37,38]. Motion cycle segmentation is another basic component for motion analysis and can be carried out by Finite State Machine (FSM), peak detection, threshold, and dynamic time wrapping (DTW)-based techniques [39,40].

### 3.2. ML-Based Methods

ML methods have been used to analyze high-volume and complex data for disease diagnosis, symptom recognition, characteristic estimation, and severity assessment. The basic idea is to map the motion data with medical diagnosis outcomes in a framework, as illustrated in Figure 5. As demonstrated in the figure, for traditional ML methods, the features are extracted manually based on human knowledge, such as the statistical magnitude, time domain, frequency domain, and symptom-specific parameters. As for deep learning models, the feature extraction and model building procedures are performed simultaneously by deep neural networks.

#### 3.2.1. Traditional ML Methods

Using traditional ML algorithms, such as artificial neural networks (ANN), decision trees (DT), support vector machines (SVM), and hidden Markov models (HMM), to solve movement detection problems has been developing for more than twenty years [41]. Among different traditional ML classifiers, SVM has proven its popularity and effectiveness by achieving the best classification performance in 20 screened studies and outperforming its counterpart classifiers in which detailed evaluation can be found [42,43,44,45,46]. The Radial Basis Function (RBF) was used as a nonlinear kernel function for SVM to improve the generalization capacity [47,48]. DT is another major approach to efficient classification, and it provides a certain interpretability that is crucial in medical applications. Models such as C4.5 and Random Forest (RF) showed competitive results in activity recognition and severity assessment [49,50,51,52,53]. Linear Discriminant Analysis (LDA), Naïve Bayesian, k-nearest neighbor (k-NN), and shallow ANN are also used as typical ML techniques to build specific classifiers. For supervised regression tasks, such as joint movement tracking or clinical score estimating, Support Vector Regression (SVR) and Gaussian Progress Regression (GPR) models are the most popular choices [54,55,56]. Some articles also used linear regression (LR) to demonstrate the relation of the medical outcomes and handcrafted features [25].

Ensemble learning is a powerful method, since it combines multiple learners through a certain strategy and usually can outperform most individual learners [57]. Besides bagging ensemble methods RF, boosting methods such as Adaboost [58], RUSBoost [59], and XGBoost [60,61] are frequently used for better accuracy and robustness.

The feature extraction methods are vital in traditional ML. For body motion monitoring, besides normal statistical and spatiotemporal features, correlation and entropy features are also used as a supplement. The basic feature groups are listed in Table 1, and the specific features for portraying each disease are introduced in Section 4. To simplify the ML model to prevent a dimension explosion, feature selection or transformation methods is the standard procedure after feature extraction. Feature selection methods can be categorized into three classes as the filter, wrapper, and hybrid methods [62]. To balance the time consumption and optimization of the model, the hybrid method that takes advantage of both the filter and wrapper methods is the most used method. It chooses feature subsets by criterion such as entropy and evaluates the subset performance by applying it to trained models [63]. For feature transformation, the Principal Component Analysis (PCA) uses an orthogonal transformation to convert raw features to compact uncorrelated new features, which is widely used to reduce the dimensions without sacrificing the accuracy [58,64,65,66].

#### 3.2.2. Deep Learning Methods

Developed from ANN, the Deep Neural Network (DNN) is more capable of learning from a large amount of data instead of utilizing handcrafted features. With the help of various network architectures, deep learning models are widely used in computer vision, speech recognition, natural language processing, and human activity recognition. For analyzing motion data captured from the human body, the convolutional neural network (CNN) and recurrent neural network (RNN) are the most used supervised deep models.

Inspired by the neurobiological model of the visual cortex [67], the CNN utilizes a series of weight-shared small filters to perform a convolutional operation over the whole input Signal, which results in two main advantages over other NN models: local dependency and scale invariance [68]. When processing the time series inertial data, local dependency means the nearby signal can be correlated, and scale invariance means the signal of different scales, but the same pattern can be invariant. After CNN, pooling and batch normalization is usually done to prevent overfitting. The output of the CNN model can be versatile to cope with different tasks. Specifically, a multi-task learning model can be built using multiple labels, as demonstrated in [69].

An RNN can be used to process the time series data by utilizing the temporal correlations between neurons. For time-related classification tasks such as the prediction of freezing of gait, RNN can learn the pattern through historical data and analyze the gait through sliding windows in real time [24]. Long Short-Term Memory (LSTM) is an improved version of RNN to conquer the long-time forgetting problem [70].

#### 3.2.3. Unsupervised Learning Methods

The accurate label of data from medicine applications can be hard to collect, as most ML methods need labels for supervised learning. The current unsupervised learning methods are useful for segmentation, clustering, and better feature extraction for further model development. HMM is a well-known directed graphical model commonly used for modeling time series data, especially for automatic gait segmentation. Using the exaptation maximization (EM) clustering algorithm and self-organizing maps (SOM) [71], the input data can be grouped through their own characteristics and provide extra feature information.

Pretrained models are the new trend in ML, and using an autoencoder to learn a latent representation of the input data for advanced feature extraction has proven to be useful and reduce the need for a large number of labeled data [72,73].

## 4. Results for Different Application Scenarios

The results of the reviewed articles in aspects of application scenarios are thoroughly described in this chapter. Each chapter is complemented with a table that provides a list of application scenarios, implementing details, and major performance (Table 2, Table 3, Table 4 and Table 5).

### 4.1. Neurological Disorders

#### 4.1.1. Parkinson’s Disease

Parkinson’s disease (PD) is one of the most common neurodegenerative diseases. It mainly causes the dystrophy of the motor system. Tremors, bradykinesia, and postural instability are the primary Parkinsonian motor symptoms, which can be monitored through IMUs attached to various body parts.

Sensors mounted on the upper extremity can be used to detect and assess the severity of tremors and bradykinesia. Dai et al. [76] proposed a method using an inertial sensor attached to the tip of the participant’s forefinger to capture finger motions. The parameters obtained by an electromagnetic tracking system (EMTS) in three tasks are used as the ground truths of the severity score from the Unified Parkinson’s Disease Rating Scale (UPDRS) of both symptoms to train common ML models. SVM achieved the best performance on three tasks and had a better-quantifying performance than neurologists. Similarly, Shawen et al. [75] used a skin-mounted IMU and a commercial smartwatch to regress the clinic-scored severity scale of tremors and bradykinesia. The Area Under the Receiver Operator Characteristic curve was used to evaluate the results, and the multiclass model of both symptoms achieved 0.74 for tremors and 0.65 for bradykinesia in the most efficient feature setting. Furthermore, the CNN was used to classify hand tremors and bradykinesia and achieved a better performance than the traditional ML methods [12,74].

Gait abnormality is another major symptom of PD patients that can be monitored by mounting IMU on the lower extremities, especially on the feet and ankles. Pérez-Ibarra et al. [78] modeled gait as an FSM and adopted an HMM-based adaptive algorithm to predict the transition state. The unsupervised model obtained a high accuracy (F1-score ≧ 0.95) using two force-sensitive resistors (FSRs) on each foot as a reference. To estimate the stride length accurately, Hannink et al. [79] adopted a two-layer CNN model to map stride-specific inertial sensor data to the resulting stride length and train the model using labels acquired from instrumented pressure mat GAITRite [4]. After various gait characteristics have been extracted, it is usually used to comprehensively assess the severity of PD. Through a series of sensitivity analyses on sensor location and feature selection, Caramia et al. [57] identified the most useful features for assessing the PD symptom severity stages per the Hoehn and Yahr (H&Y) scale. Their classification results showed that, first, the meta-classifiers based on the majority of voting achieved the best results and then, the range of motion (RoM) parameters extracted from the knee, which outperformed many spatiotemporal parameters, such as step length and step time. Rastegari et al. [58] searched for the best feature selection methods to assess the movements of PD patients and geriatrics and proved that genetic algorithms, the maximum signal-to-noise ratio with minimum correlation, and a modified version of the maximum information gain with minimum correlation are the best performers.

In advanced stages, PD patients can experience spontaneous episodes of the freezing of gait (FoG), which increases the risk of falling. Several ML strategies are adopted in detecting FoG events, such as SVM with RBF kernel [43,47], CNN [81,83], and RNN [24], all of which require expert clinicians to label using video recordings and supervised training. Compared with the detection of FoG, the prediction of FoG before the occurrence is a much more important and challenging task that requires the model to have a deep understanding of the principles and causes of the disease. Palmerini et al. [86] predicted the FoG by recognizing a degradation of the gait pattern that is significantly different from the normal gait called the pre-FoG phase. Six features were extracted from 2-s windows of the movement signals recorded by inertial sensors mounted on the shins and the lower back, and the three best features were selected to train an LDA classifier to divide the pre-FoG from normal gait.

To build a better and robust PD diagnosis and severity assessment system, movement of both the upper and lower extremities and torso can be monitored altogether using wearable inertial sensors. Using nine DOF IMUs mounted on the hands and feet, Butt et al. [70] built a Bi-LSTM model to classify 64 PD patients and 50 healthy controls, achieving an accuracy of 82.4%. To concur with the requirements of a large amount of training data of deep learning models, Som et al. [72] proposed a novel method of pretraining the model in healthy subjects performing activities of daily living (ADL) dataset to extract the relevant features for PD classification. The results showed that the autoencoder obtained a better performance than hand-engineered features in multiple sensor locations. Motor fluctuations of the PD patients between the ON state (medication working) and OFF state (medication has worn off) can be assessed in their natural environment using two IMUs [44].

#### 4.1.2. Stroke

Stroke is the second leading cause of mortality and a major cause of disability worldwide. Wearable sensor technology and ML have allowed for the seamless and objective study of human motion in the post-stroke rehabilitation progress. Predicting quantitative patient-related outcomes such as National Institutes of Health Stroke Scale (NIHSS) and the Wolf Motor Function Test (WMFT) is a major area of focus and has been covered by many articles [109,110].

Upper Extremity (UE) rehabilitation is the major application area for stroke-related IMU-ML research. Parnandi et al. [90] optimized a sensor-ML approach for clinical implementation by finding the best performance of ML algorithms and sensor configuration for classifying four different hand movement tasks. The results first showed that LDA had the highest performance on the overall positive predictive values (PPV) (92.5%) and AUCs (0.96–0.99), and then, seven IMUs on the paretic arm and trunk led to the highest classification performance.

Using wrist-worn inertial sensors, stroke-related UE motion can be automatically extracted and analyzed for severity assessment, lightening the burden of both patients and their caregivers. Oubre et al. [54] combined the unsupervised clustering algorithm and correlation feature selection (CFS) with the standard SVR, which achieves a high regression performance estimating the Fugl-Meyer Assessment (FMA) scores (normalized root mean square error of 18.2%). Reasonable motion decomposition makes the results more explainable than the other feature extraction methods but also causes a curse of dimensionality that limits deployment and application. The same idea goes when attaching a smartphone to the wrist and performing the same FMA tasks using Gradient Boosting as the classifier [61].

Some special characteristics can be estimated and used for further guidance, such as hand use capacity and arm weakness. By measuring the change in distance of the wrist and finger using IMUs, the amount of hand use can be monitored for clinicians to establish individually tailored therapeutic programs. Through the benchmark measurement of the motion capture system and an SVR-based algorithm, Liu et al. [92] established a new assessment procedure that can reveal the hand use more precisely by regressing the distance change. As for arm muscle weakness evaluation for stroke self-screening, Phienphanich et al. [50] used an RF classifier based on information gain feature selection from handcrafted features from two arm movements of the subjects. The results showed that the curl-up-only feature-based classifier achieved the most efficient results with an average accuracy of 84.1%.

Stroke also affects patients’ functional ability of their lower extremities (LE), causing partial disability and gait disorders. Inspired by deep learning techniques, Wang et al. [69] developed a two-stage DNN model for the detection of stroke gait and classification of four common gait abnormalities, including dropped foot gait, circumduction gait, hip hiking gait, and back knee gait. The developed models achieve an average accuracy of 99.35% in detecting the gait strokes and an average accuracy of 97.31% in classifying the gait abnormality. To determine the best sensor configuration for the LE function ability estimation, Derungs et al. [55] used a biomechanical simulation to synthesize the sensor reading of different positions and regression model output for the Lower-Extremity Fugl-Meyer Assessment (LE-FMA). The results showed that sensors should be preferably placed at the front of the thigh. Lucas et al. [94] proposed a method of extracting the characteristic movement patterns for the long-term monitoring of stroke recovery. The Oxford Grading Motor Scale is used for mapping the limbs acceleration data to the movement ability.

#### 4.1.3. Cerebral Palsy

Cerebral palsy (CP) is the most common physical disability among children. Inadequate physical activity (PA) is a major problem affecting the health and well-being of children with CP. ML-based approaches have the potential to improve the accuracy and versatility of inertial-based assessments of PA. Three studies [51,95,96] accomplished a series of evaluations using ML techniques to classify the PA of children with CP using IMUs mounted on the wrists, hips, and lower extremities. Surprisingly, the RF classifier proved to be effective in all three studies, achieving the best classification accuracy. The results showed that IMU-ML models may provide the accurate recognition of clinically relevant PA behaviors in children and adolescents with CP and can help clinicians monitor walking in both controlled settings and free-living situations. A gait analysis during the CP rehabilitation progress is another focus point. Chakraborty et al. [97] proposed a novel data representation method using discrete wavelet transform to form decomposed signal segments and a multi-channel one-dimensional CNN to classify abnormal gaits, achieving a better performance than the basic CNN model in the CP dataset.

#### 4.1.4. Cerebellar Ataxia

Cerebellar ataxia (CA) is a neurologic phenotype caused by a heterogeneous set of underlying diseases or injuries that affect the function of the cerebellum, causing a movement disorder of the limbs. Through features associated with accelerometric data acquired by 31 sensors located at different parts of the body, Dostal et al. [98] utilized a two-layer neural network to distinguish ataxia patients from normal subjects. The best results were achieved with an accuracy of 98.0% for sensors located in the upper part of the body (shoulders, head, and spine). Ngo et al. [99] proposed an objective framework for the diagnosing and assessment of CA based on motion data. For feature extraction, the Recurrence Quantification Analysis, Multi-Scale Entropy, Noise Effect, and Harmonic Ratio and Index of Harmonic are used to quantify the regularity or complexity in time series data from patients performing three major tests. A thorough comparison was performed over nine feature selection algorithms and 28 ML models, the Gaussian Naive Bayes classifier performed best in diagnosing CA, and the voting regression model exhibited a significant correlation (0.72 Pearson) with The Scale for Assessment and Rating of Ataxia (SARA) scores in the severity assessment in the Romberg’s test. Oubre et al. [100] made a new attempt by mapping the sub-second movement profiles obtained during a reaching task to the Brief Ataxia Rating Scale (BARS) to estimate the overall ataxia severity using GPR. Decomposed wrist movement revealed characteristics, including distance, speed, duration, morphology, and temporal relationships, that can be strongly related to disease severity and disease phenotype.

#### 4.1.5. Others

There are other neurodegenerative disorders, such as Huntington’s disease (HD), Progressive Supranuclear Palsy (PSP), and Multiple Sclerosis (MS), that have benefitted from recent research on ML-based inertial sensors for characteristics extraction, activity monitoring, assistive diagnosis, and objective assessment. Bennasar et al. [101] proposed an accelerometer-based quantitative assessment of upper limb movement impairment for HD patients. An ensemble classifier and a LR model are used to first classify HD patients and healthy participants and then generate a continuous movement impairment score that correlates with the clinical HD rating scale, achieving a 98.78% accuracy in the classification and Pearson correlation coefficient of 0.77, with a *p*-value < 0.01. For the early and accurate diagnosis of PSP, which may be difficult to distinguish from idiopathic PD, De Vos et al. [102] used the RF classifier and 17 features to discriminate PSP from PD, achieving 86% sensitivity and 90% specificity. Fatigue, the most common symptom in MS, can also be predicted using an ankle-mounted IMU and an RF classifier [64].

Research on the monitoring and rehabilitation of patients who suffered from neurological trauma, such as traumatic brain injury (BI), spinal cord injury (SCI), and brachial plexus injury (BPI), have also produced results. For monitoring the recovery process for stroke and traumatic brain injury survivors, Lee et al. [56] proposed a GPR-based regression model to estimate rehabilitation outcomes using a combination of clinical and wearable inertial sensor data. This approach resulted in a Pearson’s correlation of 0.94 between the estimated and clinician-provided scores. Wearable sensors can also be used to quantify shoulder dysfunction following brachial plexus injury, as shown in [103]. Nazarahari et al. compared the bilateral asymmetry of the six calculated kinematic scores of the affected and controls and used a bagged ensemble of decision trees for classification. The results showed a significant difference in the asymmetry indexes, concluding a promising classification accuracy.

Functional disorders such as seizure and vestibular system disorder and neuromuscular disorders such as peripheral neuropathy can be diagnosed through ML approaches [62,104,105]. Ikizoglu et al. [62] compared two-dimensional reduction techniques, including feature selection and feature transformation for SVM, to identify Vestibular System (VS) disorders. As the most used feature transformation method, kernel-modified PCA achieved the best performance, with a 89.2% classification accuracy. Finally, there are general movement assessment methods such as gait assessment that diagnose symptoms and benefit from multiple neurological disorders [106,107]. Other symptoms observed in neurological impairment patients such as spasticity can also be assessed by estimating clinical scales such as the modified Ashworth scale [108].

**Table 3 healthcare-10-01210-t003:** Musculoskeletal disorders.

Disorders	Application	Sensor (n)	Placement	Model	Input Data/Features	Major Performance	Subjects/ Dataset	Year	Ref.
OA	SD	IMU (2)	32, 33	ANN	100 samples	multiple	14 h	2020	[111]
OA	SD	IMU (1)	31	LR	63 features	MAE = 29% (left), 36% (right)	10 p	2020	[25]
OA	CE	Accel (1)	31	RF	26 features	Acc = 76.3%	1198 p [112]	2021	[53]
OA	CE	Accel (3)	2, 13, 35	SVM	temporal features	Acc = 97.9% (initial), 90.6% (layer-1 SVM), 92.7% (layer-2 SVM)	10 h	2016	[113]
OA	SD	Accel (4)	13, 16, 29, 35	LDA + PCA	38 features	Acc = 81.7%	39 p	2017	[65]
OA	PA	IMU (4)	23, 24	CNN	200, 100, 40 ms window	Acc = 85%, 89–97%, 60–67% for 3 tasks	18 p	2021	[114]
LBP	D	IMU (1)	2	SVM/MLP	16 features	Acc = 75%	94 p	2020	[115]
LBP	D	IMU (2)	2, 11	SVM	52 features	Acc = 96%	28 p, 24 h	2017	[48]
TJR	D	IMU (7)	11, 23, 26, 34	SVM	2 feature sets	Acc = 87.2% (Set 1), 97.0% (Set 2)	20 p, 24 h	2019	[46]
TJR	PA	IMU (4)	13, 14, 16, 29	DCNN	100 samples	Acc = 98%	12 p	2021	[116]
TJR	SA	Accel (2) IMU (1)	6, 11	k-means	Different for each PROM	TSS = 3.86, 3.56, 1.86 for each feature set	22 p	2019	[117]
TJR	PA	IMU (1)	14	SOM	356 features	Acc = 85.6–96.92%	44 p, 10 h	2018	[118]

### 4.2. Musculoskeletal Disorders

#### 4.2.1. Osteoarthritis

Osteoarthritis (OA) is the most common musculoskeletal disease and will be diagnosed in nearly half of all people at some point in their life [65]. Specifically, knee OA (KOA) accounts for more mobility disabilities in people over the age of 65 than any other medical condition. Joint movement measurements, especially knee flexion moments (KFM) and knee adduction moments (KAM), represent an objective parameter of the knee joint load in KOA. Stetter et al. [111] developed an ANN that estimates the formerly mentioned parameters based on time series data obtained by two IMUs located on the right thigh and shank. For all six locomotion tasks, the ANN achieved a high overall concordance in KAM (r = 0.39 ± 0.32, rRSME = 29.9 ± 8.1%) and KFM (r = 0.74 ± 0.36, rRSME = 20.8 ± 5.7%), which is essential for KOA patients to provide valuable biofeedback systems. Joint loading is also valuable for hip OA patients and can be monitored by a mobile phone attached to the patient’s hip. Brabandere et al. [25] proposed an ML pipeline for learning the musculoskeletal modeling to estimate the loading value using only an embedded IMU, and the proposed LR-based pipeline achieves a mean absolute error of 29% for the left hip and 36% for the right hip. Sun et al. [53] proposed an ML model to bridge the motion features and thresholds of KOA patients with longitudinal gait decline to estimate the personal physical capacity and found that the most impactful predicting feature is low minutes during the performance of moderate-vigorous activity. Rehabilitation exercise plays an important role in KOA therapy. Motion segmentation is the main difficulty in rehabilitation monitoring. Chen et al. [113] proposed a multi-layer SVM-based online segmentation model and achieved a segmentation accuracy of 92.7%. To predict a performance improvement in muscle-strengthening exercises, Kobsar et al. [65] established an ML-based grading system that can predict the post-intervention response to an exercise therapy through preintervention multi-sensor accelerometer data. Overall, the best performance of classifying different responders is 81.7% accuracy, which is achieved by sensors mounted on the back, thigh, and shank.

#### 4.2.2. Low Back Pain

Low back pain (LBP) is expected to substantially increase, along with the world’s aging population, due to the normal physiological intervertebral disc and tissue degeneration associated with aging [48]. Using IMU mounted on the sternum of patients, Abdollahi et al. [115] managed to predict LBP by classifying patients into high-risk and low-risk subgroups. Accuracy levels of ~75% and 60% were achieved for the SVM and MLP, respectively. Using a similar sensor configuration, Ashouri et al. [48] proposed an approach for evaluating LBP in various settings. The results showed that, through a low-pass filter, PCA for feature extraction, and SVM for classification, the model can be adequately employed for LBP identification, with an accuracy of 96%.

#### 4.2.3. Total Joint Replacement

Patients after undergoing TJR surgeries suffer from lingering musculoskeletal restrictions. Teufl et al. [46] combined ML approaches with the gait analysis to classify these impairments. Features such as spatiotemporal parameters and joint angles are proposed as the input for a SVM model, achieving an accuracy of 97.0%. DCNN is employed for monitoring the progress of the rehabilitation of hip unilateral arthroplasty surgery by mapping the sensor’s gait cycles data to days after surgery [116]. The proposed DCNN achieved up to 98% classification accuracy for the rehabilitation progress monitoring. Bini et al. [117] used data from three commercially available wearable sensors and an ML algorithm to predict the postoperative clinical outcome scores of TJR patients. The K-means ML algorithm was used to cluster the patients into three groups according to the variables that were more correlated with the clinical outcomes. The quantitative variables (steps taken, heart rate variability, and calories burned) were found to have better predictive power than the qualitative ones (cadence, bounce, and braking) for the Rand 12-item Health Survey scores and qualitative scores for the Hip Disability and Osteoarthritis Outcome Score and Knee Injury and Osteoarthritis Outcome Score surveys.

**Table 4 healthcare-10-01210-t004:** Mental illnesses.

Disorders	Application	Sensor (n)	Placement	Model	Input Data/ Features	Major Performance	Subjects/ Dataset	Year	Ref.
Depression	D	Accel (1)	6	RF	14 features	Acc = 89.2%	2112 p, 3783 h	2019	[119]
Depression	D	Accel (1), Light	6	Logistic Regression	4 features	Acc = 91%	18 p, 29 h	2019	[120]
Depression	D, SA	Accel (1), Health	6	XGBoost	63 features	Acc = 76%, correlation coefficient = 0.61	45 p, 41 h	2020	[121]
Depression	D	Accel (1)	6	RF	3 features	MCC= 0.44	23 p, 32 h	2018	[122]
Depression	SA	Accel (1)	6	RF, Adaboost, Theil-Sen	3 sets of features	RMSE = 4.5	12 p	2017	[123]
Bipolar, ADHD	D	Accel (1)	11	SVM	28 features	Acc = 83.1%	92 p, 63 h	2016	[124]
Internalizing Disorders	D	IMU (1)	11	Logistic Regression	39 features	Acc = 81%	21 p, 41 h	2019	[125]

### 4.3. Mental Illness

Mental Illness is an umbrella term that describes disorders that affect people’s emotions, thoughts, and behaviors. Mental illness is typically diagnosed qualitatively—patients come to psychiatric professionals with symptoms, and the professionals diagnose them using inquiries, tests, and observations of behavior. Then, they start treating the patients with psychotherapy, medication, or both [126]. Monitoring the effectiveness of the treatment is critical in mental illnesses, as adjustments in medication might be necessary along the way. Yet, physicians rely on patients for input in a treatment effectiveness and even waiting and seeing if a particular dose is effective or not [127]. Using wearable sensors that monitor the physiological measures in real time would allow physicians to check up more frequently on their patients and adjust their treatments accordingly.

#### 4.3.1. Depression

Depression symptoms have different categories, such as irregular sleep, change in daily life activities, mood changes, and overeating or not eating enough. Many studies have tracked these symptoms to predict depression outcome scores. IMUs were specifically used to track the activity levels.

Zanella Calzada et al. [119] used motor activity data from a wrist-worn Actigraph to detect depression. The classification of the control and depression according to these selected features was done by using RF. The validation results show that 89.2% of participants that had depression were correctly classified by their algorithm. Kim et al. [120] also used Actigraph to predict depression, this time in older adults. They also compared the performance of four different ML methods: logistic regression, DT, boosted trees, and RF. Among these, logistic regression performed the best, with a 91% accuracy. Tazawa et al. [121] reported lower accuracy levels of the XGBoost technique, 76% percent for identifying depression patients and 61% for predicting depression outcomes. In one study, Garcia-Ceja et al. [122], unlike the other studies presented here, measured the activities of unipolar and bipolar patients directly to detect depression. RF outperformed the DNN in classifying depression with an MCC value of 0.44 compared to 0.39. This group also compiled a dataset [128] of these movement recordings.

Other studies utilized a combination of wearable sensors and smartphones to detect depression. Ghandeharioun et al. [123] used wrist-worn sensors and smartphones to track depression symptom categories to predict the scores for the Hamilton Depression Rating Scale (HDRS). They used a combination of different ML methods such as Adaboost, RF, and Theil-Sen. They managed to achieve a low RMSE rate of 4.5% in predicting HDRS scores.

#### 4.3.2. Other Mental Illness

Although depression studies make up the majority of studies that take advantage of the IMU-ML approach, there are other mental illnesses such as bipolar disorder, attention deficit disorder (ADD), and schizophrenia. Faedda et al. [124] used Actigraph to distinguish two different mental health disorders, bipolar and attention-deficit/hyperactivity disorder (ADHD) in children and adults. Among the five ML methods they tested, SVM had the highest performance, with 83.1% accuracy. McGinnis et al. [125] also used wearable sensors and ML to detect mental illness—in their case, in children. A belt- worn IMU was worn by the children as they underwent a “Snake Task” that elicits fear. The task is made up of three parts: potential threat startle, response modulation, and the researcher reassures the child that the snake is fake. There were 29 features extracted from the IMU for each of the six time series (two for each part of the task). The classification results were validated using three methods: logistic regression with a score threshold for accuracy, specificity, and sensitivity; AUC; and a permutation test to determine the results that were obtained by chance. Among the three parts, the potential threat was found to be the best measure for detecting mental illness in children.

**Table 5 healthcare-10-01210-t005:** Other disorders and general rehabilitation.

Disorders	Application	Sensor (n)	Placement	Model	Input Data/ Features	Major Performance	Subjects/ Dataset	Year	Ref.
COPD	SA	Acc (1)	-	SVM_rbf	8 features	Acc = 99.2%	55 p, 11 h	2016	[129]
COPD	CE	IMU (3)	2, 11, 30	PCA	Quaternion data	MAE < 2, R > 0.963	8 h	2019	[66]
Geriatrics	D	IMU (1)	31	CNN+LSTM	500 samples	Acc = 95%	20 p	2021	[130]
General	CE	IMU (2)	13, 35	Polynomial Regression	Orientation	RMSE = 4.81 (general),	14 h	2019	[131]
General	PA	IMU (4)	4, 5, 6, 7	RF	2 feature sets	4.99 (personal)	50 h	2020	[132]
General	PA	IMU (1)	5	RF/SVM	237 features, ReliefF	Acc = 98.6%	44 p, 10 h	2019	[133]
General	PA	IMU (3)	3, 5, 6	Conv+FSM	Raw	Acc = 97.2% (CV), 80.5% (LOSO)	35 h	2020	[134]
General	PA	IMU (2)	5, 6	SVM	144 features, PCA	Acc = 0.871	9 p, 9 n	2021	[135]

### 4.4. Others

There are other diseases such as pulmonary diseases [129], geriatrics [130], and chronic pain [133] that can benefit from objective assistive monitoring and assessment. Chronic obstructive pulmonary disease (COPD) is a common progressive disease affecting the airways, causing chronic morbidity [129]. Cesareo et al. [66] utilized three IMUs attached to the chest of patients to monitor the breathing frequency, which is a vital sign for the breathing function assessment. PCA fusion of the four quaternion data provided the best performance in terms of the MAE (<two breaths/min).

The monitoring of general human movement, including upper extremity, lower extremity, joint angle, and physical activity during rehabilitation, can be versatile to cope with different disease requirements [131,132,133,134]. Especially, IMUs can provide an accurate and reliable method of a joint angle assessment with the application of ML algorithms. Argent et al. [131] compared four major regression models and utilized an IMU platform placed on the lower extremity and a 3D motion capture system for the gold standard to train the model. The use of an orientation algorithm as a preprocessing step improved the accuracy, and the average RMSE for the best-performing algorithms, orientation polynomial regression (PR), across all exercises was 4.81°.

## 5. Discussion and Future Directions

Using IMUs to monitor human motions for disease-specific characteristics and biomechanical applications has been around for years and is developing rapidly with the maturity of the sensors and processing units and more and more sophisticated algorithms, especially ML algorithms. Plenty of successful attempts, which allow medical staffs to acquire a patient’s body information remotely, have been observed in IoHT scenarios. Although ML has brought great success to the research in several aspects, there are still challenges for IMU-based body motion monitoring.

### 5.1. Inertial Sensors and IoT Devices

IMU attached to the human body is expected to generate real-time data that dynamically represent the movements of a human body (or a part of it). However, due to the IMU sensors’ intrinsic design-induced side effects (such as offsets, zero drift, and nonorthogonal error), together with environmental inference (e.g., EMI and thermal noise), the output motion data normally present low SNR values and distortions. Although various calibration techniques have been proposed [20,21,136], obtaining high-quality data is still challenging, which results in insufficient information for ML models to interpret the patients’ motion status. Furthermore, the current IMU modules are bulky, not only affecting the natural movement of users but also limiting the continuous monitoring of daily activities.

#### 5.1.1. Multi-Sensor Fusion

To surpass the limitation of sensors, multi-sensor fusion is the most used method for medical applications. Some variants of IMU, such as MARG (Magnetic, Angular Rate, and Gravity), which incorporates a tri-axis magnetometer, can provide a complete measurement of the orientation relative to the Earth’s magnetic field. To capture the source of motion for humans, a surface electromyography (sEMG) sensor is usually applied as it directly reflects the neural information when flexing the muscles [137]. A mechanomyography (MMG) sensor is frequently used as the “mechanical counterpart” to an EMG and is formed from pressure waves to provide more accurate myographic information [138]. FSRs integrated into shoes and insoles are utilized to measure the force in various applications, such as accurate gait segmentation, Center of Gravity (CoG) estimation, and detailed gait parameter estimations [139].

#### 5.1.2. Self-Calibration

Self-calibration approaches in the inertial sensors have been developed to address the long-term drift problem. There exist many sensing structures or mechanism innovations for in situ self-calibration that can optimize the characteristics of specific inertial sensors [140,141]. For example, continuous calibration can cancel drive-induced errors in gyroscopes without interrupting normal operations and affecting the noise and bandwidth by periodically reversing the polarity of the gyroscope’s forcing voltage [142]. For IMU platforms, self-calibration can be performed by using a more sophisticated error model and a set of motion procedures to capture the correlations between sensors [143,144]. Through the development of the sensor platform and the portrayal of the error model, the sensors can be more accurate and reliable in long-term monitoring.

#### 5.1.3. Consumer Grade IMU Devices

Although IMUs are much more affordable than other motion capture methods, the most commonly used IMUs in research are still inaccessible to the general public. The two most commonly used IMUs in research, Xsens Awinda [145] and Shimmer [146], still cost thousands of dollars and require dedicated training to operate. However, there are many IoT devices that record movements that can be a viable alternative to lab grade equipment. Already, some studies are testing the viability of using consumer products for rehabilitation. Additionally, these consumer grade products will allow for longer monitoring times and even the detection of illnesses beforehand through ML-based prediction methods informing physicians.

### 5.2. Data Processing Methods

To sort out the context of the reviewed articles based on the two criteria of this review, we summarized the data processing methods and detailed the application scenarios, as shown in Figure 6. The data processing methods were composed of traditional machine learning, deep learning, and unsupervised learning methods, as demonstrated in Section 3.2. Detailed application scenarios were clustered into:Disease Diagnosis, which means classifying patients from healthy controls;Symptom Detection, which means detecting a typical symptom of patients that indicates a detailed disease type and stages, such as the freezing of gait for PD patients;Characteristics Estimations, which means estimating disease-related characteristics such as stride length and joint loading value;Severity Assessment, which means regressing or classifying different severities of patients into certain estimating scales;Physical Activity Recognition for Patients.

The ML methods of the reviewed articles are categorized and shown as bubbles in a given cell of the data processing method (row) and detailed application scenarios (column). Since some articles covered multiple application scenarios, the corresponding methods were counted more than once. The color of each bubble represents the number of reviewed works covering a given cell. In that way, we can locate the commonly used methods and a few optional choices for the target application to accelerate the implementation process.

Although this review has shown a series of reliable IMU-ML systems, there still exists a considerable gap between the technical tools and clinically usable systems, since most of the existing system is built in a lab to collect patients’ data and process them afterward. As urgently needed in many real-time IoHT scenarios, such as telerehabilitation, at-home diagnosis, and biofeedback systems, the deployment of ML algorithms for real-time streaming data in real-world settings is still a pending issue. For building an accurate and robust ML model for diagnosis and assessment, large and comprehensive datasets with accurate labels are needed. Since different diseases could share common characteristics and symptoms, and sometimes a patient may even suffer from multiple diseases, a general model for tracking a multitude of diseases instead of specific models would provide more utility. The model should have the ability to provide and explain useful information to the clinicians, such as the meaning of extracted features, reasoning process, and the most valuable factors to support the output results. However, most of the existing methods use ML models as a black box, lacking explanation, which causes patients and physicians to be doubtful of the reliability of the model.

#### 5.2.1. Online and Edge Implementation

To build a real-time monitoring system, the data processing model has to be deployed online or on an edge platform. Since the model is usually trained offline on a dataset in laboratory settings, and the existence of environmental noise in the real-world, incremental learning can be a useful tool to diminish the distribution shift. For latency-sensitive applications such as urgent medical cases, real-time optimization is needed to keep the balance between accuracy and time cost [6]. There are two approaches to tackle this problem: bringing up the response time as a critical element of model training and reducing the time cost during model predicting by reducing the communication cost between the sensing device and server or enhancing the computing ability of the edge devices.

#### 5.2.2. Open Dataset and Universal Model

Some of the existing works are based on open-source datasets such as Rampp et al.’s [36] dataset for gait assessment, the Cupid dataset [131] for FoG detection, Guerra et al.’s [89] dataset for post-stroke UE assessment, and the Osteoarthritis Initiative dataset [53] for the physical performance estimation. However, most of the previous works have used self-collected data limiting the development of a better and unified model. The open datasets annotated with ground truth need to be established in the field of inertial-based motion monitoring. Unsupervised and weakly supervised learning can mitigate the need for accurate labels. Pretraining and few-shot learning can be the solution by learning the human motion representation from a mass of ADL data and finetune the model using the data collected from patients.

#### 5.2.3. Interpretable Model

Considering the accountability and transparency nature of the medical applications, the interpretability of ML models for assistive diagnoses is crucial. Oubre et al. [100] showed a promising direction by the decomposition of sophisticated motions. The classification results could be explained through the contribution of different feature components that lead to the cause of disease. Morrison et al. [147] utilized visualization to support the decision-making process in a way that allowed clinicians to integrate the algorithm’s result into their decision process. For deep learning models, the explanation can be difficult, since the motion data is fed directly into the model. The development of global explanations, such as kernel-based methods [148] and Local Interpretable Model-agnostic Explanations (LIME) [149], can be useful for explaining deep models. Using the interpretable ML models, a human–AI collaborative decision-making system can be established with better fault tolerance and human rights friendly.

#### 5.2.4. Healthcare Representation and Digital Twin

Through continuous multi-sensor data collected from patients, electronic health records (EHR) can be modeled using graph-based methods and representation learning [150]. Learning meaningful medical ontology representations within the EHR database can alleviate the data insufficiency problem, and the learned embeddings can cluster nicely into particular groups of diseases [151]. With simulation and motion data synthesis, digital twin models can be established for patient performance estimation and rehabilitation program planning to individual needs [152,153]. The twin model can serve as a counterpart of the patients to improve the wearability of sensor systems and transfer detailed human movement characteristics to real-world applications to reduce errors.

## Figures and Tables

**Figure 1 healthcare-10-01210-f001:**
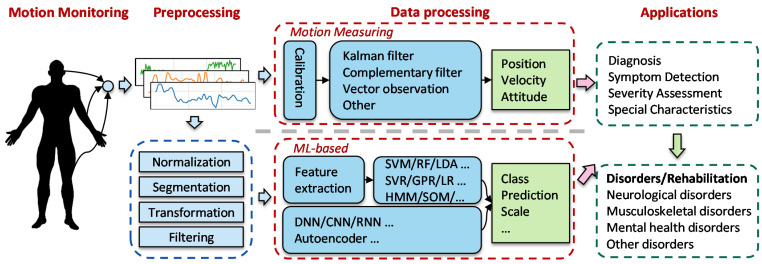
Generalized workflow for monitoring human body motions.

**Figure 2 healthcare-10-01210-f002:**
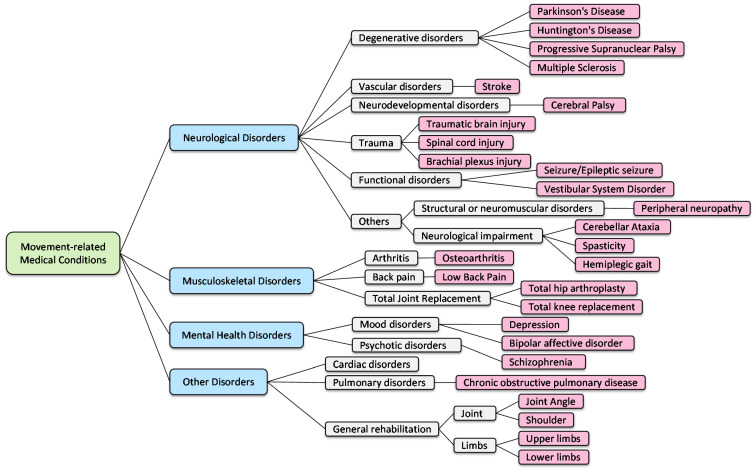
A taxonomy of the selected works.

**Figure 3 healthcare-10-01210-f003:**
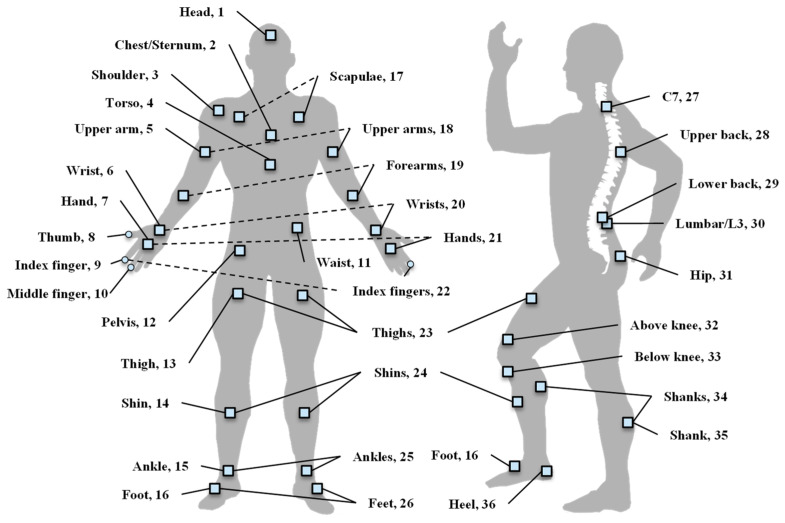
Distribution of the sensor locations.

**Figure 4 healthcare-10-01210-f004:**
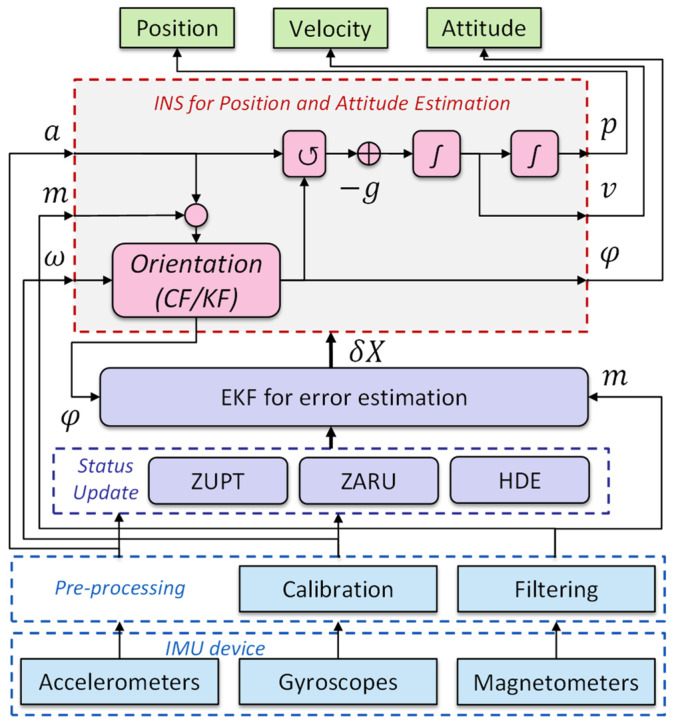
Traditional inertial tracking system. *a*: acceleration; *m*: magnetic field; *ω*: angular rate; *p*: position; *v*: velocity; *φ*: attitude; *δX*: estimated error. Abbreviations: ZARU: Zero Angular Rate Update; HDE: Heuristic Drift Elimination.

**Figure 5 healthcare-10-01210-f005:**
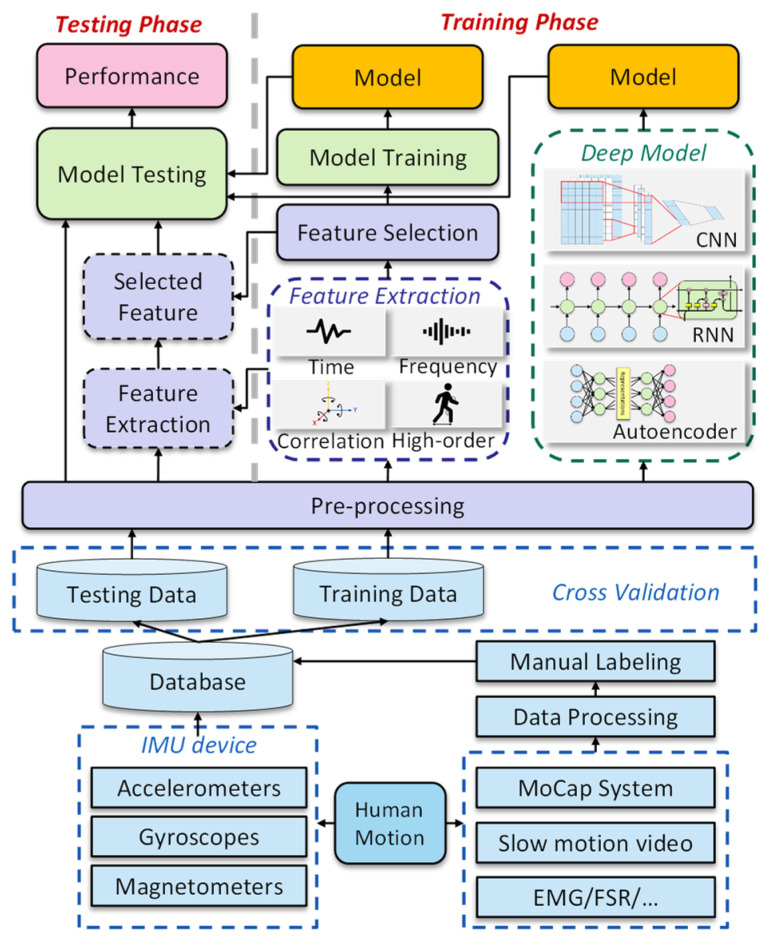
Machine learning-based method.

**Figure 6 healthcare-10-01210-f006:**
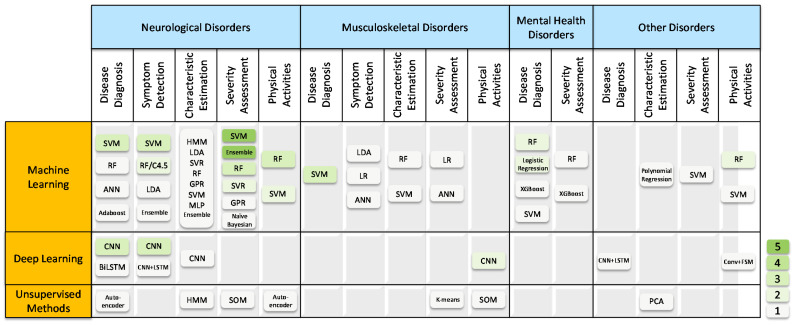
A summary of the reviewed works. The color of each bubble represents the number of reviewed works covering a given data processing method (row) and detailed application scenarios (column).

**Table 1 healthcare-10-01210-t001:** Feature groups for supervised machine learning models.

Feature Categories	Features
Time	Standard deviation, mean, range, amplitude, root mean square, variance, skewness, kurtosis, coefficient of variation (CV), increment, power, energy, and jerk Segment time, zero-crossing ratio, number of peaks DTW coefficient, and autoregression coefficient
Frequency	Dominant frequency, power of dominant frequency, amplitude in certain bandwidth, moments of power spectral density, CV of frequency, and relative magnitude
Entropy	Sample entropy, spectral entropy, and approximate entropy
Correlation	Cross-correlation (peak and lag); autocorrelation (peaks, number, sum, amplitude, and lag)
High-order	Velocity, stride/step length, left and right asymmetry, range of motions, freezing index, and harmonic ratio

**Table 2 healthcare-10-01210-t002:** Neurological disorders. The numbers in the Placement column are demonstrated in Figure 5. Abbreviations: Accel: Accelerometer; Gyro: Gyroscope; D: Diagnosis; SD: Symptom Detection; SA: Severity Assessment; CE: Characteristics Estimation; PA: Physical Activity; h: healthy controls; p: patients.

Disorders	Application	Sensor (n)	Placement	Model	Input Data/Features	Major Performance	Subjects/ Dataset	Year	Ref.
PD	D	IMU (1)	6	CNN	28 samples	Acc = 97.32%	5 p, 5 h	2021	[74]
PD	SD	IMU (2)	19	CNN	5s window	Acc = 90.9%	10 p	2016	[12]
PD	SD, SA	IMU (2)	6, 7	RF	74 features	multiple	13 p	2020	[75]
PD	SA	IMU (1)	9	SVM	7 features	Acc = 96–97.33%	45 p, 30 h	2021	[76]
PD	SD, SA	IMU (2)	6, 15	XGBoost	78 features	R = 0.96 (ho), 0.93 (loso)	24 p	2019	[60]
PD	CE	IMU (2)	26	HMM	raw	G < 0.25	26 p, 11 h	2018	[77]
PD	CE	IMU (2)	16, 36	HMM	raw	F1 ≥ 0.95	7 p, 5 h	2020	[78]
PD	CE	IMU (2)	16, 36	CNN	256 samples	acc. ± prec. = 0.01 ± 5.37 cm	116 p [36]	2018	[79]
PD	SA	IMU (8)	2, 23, 24, 26, 30	Meta-classifier	18 feature sets	Acc = 84.00% ± 6.54%	25 p	2018	[57]
PD	D	IMU (2)	25	Adaboost	21 gait features	Acc = 85–95%	20 p,10 h [80]	2020	[58]
PD	SA	IMU (6)	6, 8, 9, 10, 26	SOM	41 features	Acc = 95% (2 classes), 81.7% (3 classes)	30 p	2019	[71]
PD	SA	IMU (4)	20, 25	SVM	178 features	R = 0.93, (0.85 (dys.), 0.84 (brady.), 0.79 (gait))	19 p	2020	[42]
PD	SA	IMU (5)	20, 25, 29	RUSBoost	134 features	AUC = 0.76–0.90, Sen = 72–83%, Spec = 69–80%	332 p, 100 h	2021	[59]
PD	SA	Accel (1)	29	SVM	temporal features	Acc = 92.3%, 89.3%, 85.9 for 3 binary classifications	99 p, 38 h	2016	[81]
PD	SD	IMU (3)	24, 29	SVM_rbf	86 features	Acc = 85.0%, Sen = 84.1%	71 p	2020	[47]
PD	SD	IMU (3)	25, 27	CNN	4s window	Acc = 89.2%	67 p	2020	[82]
PD	SD	Accel (3)	14, 15, 31	CNN	2–5s window	Sen = 93.44%, Spec = 87.38%	10 p [83]	2020	[84]
PD	SD	Accel (1)	29	SVM	55 features	GM = 76.8%, 84.0% (personal)	21 p	2017	[43]
PD	SD	Accel (1)	11	CNN + LSTM	4 features	AUC = 0.936	21 p [43]	2020	[24]
PD	SD	Accel (1)	30	C4.5	2 feature sets	Acc = 82.7%, 77.9% (2 modes)	12 p	2020	[49]
PD	SD	IMU (3)	24, 29	LDA	8 features	AUC = 0.76, Sen = 0.84	11 p [85]	2017	[86]
PD	D	IMU (6)	6, 8, 9, 10, 26	BiLSTM	190 features	Acc = 82.4%	64 p, 50 h	2020	[70]
PD	PA	Accel (6)	2, 20, 25, 30	Autoencoder	250 samples	F1 = 73.89 ± 5.69	18 p, 16 h [87]	2020	[72]
PD	SD	Gyro (2)	6, 15	SVM	3 feature sets	Acc = 83.56%	19 p	2020	[44]
PD	D	Accel (3)	4, 20	Autoencoder	1s window	AUC = 0.77	[83], 6 p [88]	2018	[73]
Stroke	CE	IMU (11)	1, 2, 12, 17, 18, 19, 21	LDA	statistical features	Acc ≥ 93%	10 h, 6 p [89]	2019	[90]
Stroke	SA	IMU (2)	2, 6	SVR	109 features	RMSE = 18.2%, R = 0.70	36 p, 32 h	2020	[54]
Stroke	SA	IMU (1)	6	SVM	statistical features	Acc = 97.70%	20 p	2019	[91]
Stroke	SA	IMU (1)	6	XGBoost	SMA feature	Acc = 95.56%	10 p	2020	[61]
Stroke	CE	Accel (4)	20, 22	SVR	271 features	nRMSE = 0.11, R = 0.78	10 p, 10 h	2019	[92]
Stroke	CE	IMU (1)	7	RF	3 feature sets	Acc = 84.1%, Sen = 94.8%	7 p	2020	[50]
Stroke	D	IMU (2)	25	DCNN	gait cycle	Acc = 99.35% (detection),	30 p, 15 h	2021	[69]
Stroke	SA	Accel (1)	13	SVR	20 features	97.31% (classification)	8 p [93]	2019	[55]
Stroke	SA	Accel (4)	19, 24	SVM	9 features	nRMSE = 0.32% (affected), 0.36% (unaffected)	18h	2019	[94]
CP	PA	Accel (3)	6, 15, 31	RF	15 features	*p* < 0.05	38 p	2020	[95]
CP	PA	Accel (2)	6, 31	SVM	27 features	Acc = 99.0–99.3%	22 p	2018	[96]
CP	PA	IMU (3)	6, 13, 31	RF	40 features	Acc = 82.0–89.0%	11 p	2020	[51]
CP	D	IMU (2)	13, 14	CNN	120 samples	Acc = 92%	9 p, 9 h	2020	[97]
CA	D	Accel (6)	1, 3, 13, 14, 16, 27	ANN	DFT features	AUC = 0.98	25 p	2021	[98]
CA / PD	SA	IMU (2)	15, 28	Naive Bayes	6 feature sets	Acc = 77.1%, 78.9%, 89.9%, 98.0%, 98.5% for 5 places	62 p, 24 h	2021	[99]
CA	SA	IMU (1)	6	GPR + GPC	53 features	Acc = 88.24%	88 at, 44 pd, 34 h	2021	[100]
HD	SA	Accel (3)	2, 20	Meta-classifier	234 features	RMSE = 3.6, R = 0.69	234 features	2018	[101]
PSP	D	IMU (6)	2, 20, 26, 30	RF	17 features	Acc = 98.78%, R = 0.77, MAE = 12.41%	21 psp, 20 pd, 39 h	2020	[102]
MS	SA	IMU (1)	15	RF	6 gait features	Sen = 86% (PSP/PD),	49 p	2020	[64]
BI	PA	Accel (1)	32	RF	statistical features	90% (PSP/HC)	25 p, 11 h	2021	[52]
SCI	PA	Accel (1)	11	SVM	temporal features	MAE = 1.38	13 p	2017	[45]
BI/Stroke	CE	Accel (5)	2, 5, 6, 8, 9	GPR	temporal features	Sen = 88.3–90.4%	44 p	2021	[56]
BPI	CE	IMU (3)	2, 18	Ensemble	20 features	Acc = 91.6%, 85.9% (at home)	15 p, 15 h	2021	[103]
Seizure	D	Accel (4)	20, 25	LS-SVM	140 features	RMSE = 6.9%, R = 0.94	51 p	2017	[104]
VS	D	IMU (5)	11, 23, 26	SVM	22 features	Acc = 93%, R = 0.55–0.76	16 p, 21 h	2020	[62]
General	D	IMU (2)	34	SVM	8 gait features	multiple	36 p, 13 h	2020	[105]
General	CE	IMU (4)	23, 24	SVM	16 gait features	Acc = 89.2%	25 p, 24 h	2017	[106]
General	CE	IMU (1)	6	MLP	statistical features	Acc = 93.9%	10 p	2019	[107]
Spasticity	SA	IMU (1)	6	RF	2 feature sets	Acc = 91.61%	50 p	2020	[108]

## Data Availability

Not applicable.

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
