# Peer review of "IMU-Based Monitoring for Assistive Diagnosis and Management of IoHT: A Review"

_healthcare, 2022, doi:10.3390/healthcare10071210_

Round 1
Reviewer 1 Report
1. The authors talk about 3 categories at the bottom of page 7, but name 4 groups in the rest of the paragraph. It is recommended to revise the text of this section.
2. authors can also refer to problems such as big data and real-time decision making or challenges of dynamic environments/systems and streaming data processing which are important issues about IOT, in Discussion and Future Directions section.
3. Please be sure about the claim made in Section 2.2.1 ("SVM is the most popular classifier and outperforms its counterparts"), as the references mentioned do not indicate comprehensiveness
Author Response
Response to Reviewer 1 Comments
Point 1: The authors talk about 3 categories at the bottom of page 7, but name 4 groups in the rest of the paragraph. It is recommended to revise the text of this section.
Response 1: We have revised the text of this section and changed the number 3 to 4.
Point 2: authors can also refer to problems such as big data and real-time decision making or challenges of dynamic environments/systems and streaming data processing which are important issues about IOT, in Discussion and Future Directions section.
Response 2: We have emphasized the challenges of building a real-time system in the 5.2 data processing methods part of the Discussion and Future Directions section. We have also added a paragraph introducing the foreseeable directions to tackle this problem named “5.2.1 Online and edge implementation”.
Point 3: Please be sure about the claim made in Section 2.2.1 ("SVM is the most popular classifier and outperforms its counterparts"), as the references mentioned do not indicate comprehensiveness
Response 3: We have rephrased the claim and provided the reason for SVM being the most popular traditional ML method. We have also provided a summarization figure to show the popular model choice of different application scenarios in figure 6.
Reviewer 2 Report
I am not a specialist in the field of IoHT, but a very interested reader. So that I have read this review paper immediately, with enthusiasm, upon receiving the invitation. I have found this work, clearly organised, readable, informative and useful to a general readership. I would only propose an extra effort to the authors that could increase the impact of their review article. Making reference to figure 4 it would be nice to have a table collecting typical specificities and sensitivities (or other measures of performance, such as accuracy) in the prediction/classification of the main categories of disorders here investigated: Neurological, Musculoskeletal, Mental, Others. Indeed, in many cases this information is given in the test, but perhaps collecting all the figure in one table could give an overall idea of the state of the art in the whole field.
Author Response
Point 1: I am not a specialist in the field of IoHT, but a very interested reader. So that I have read this review paper immediately, with enthusiasm, upon receiving the invitation. I have found this work, clearly organised, readable, informative and useful to a general readership. I would only propose an extra effort to the authors that could increase the impact of their review article. Making reference to figure 4 it would be nice to have a table collecting typical specificities and sensitivities (or other measures of performance, such as accuracy) in the prediction/classification of the main categories of disorders here investigated: Neurological, Musculoskeletal, Mental, Others. Indeed, in many cases this information is given in the test, but perhaps collecting all the figure in one table could give an overall idea of the state of the art in the whole field.
Response 1: Thank you for your kindly advice! And we have added the model performance provided by each reviewed article into our tables (i.e. tables 2, 3, 4, 5) for a more obvious comparison. Since the experimental settings of each article is different, and in every detailed scenario performance of different models can be diverse. We recommend not to compare each article directly, but to use it for reference and overall guidance towards a better technical implementation. We have also provided a summarization figure (figure 6) to show the popular model choice of different application scenarios.
Reviewer 3 Report
With the rapid development of Internet of Things (IoT) technologies, traditional disease diagnosis carried out in medical institutions can now be performed remotely at home or even ambient environments, yielding the concept of Internet of Health Things (IoHT).
Among the diverse IoHT applications, inertial measurement units (IMUs) based systems play a significant role in the detection of diseases in many fields, such as neurological, musculoskeletal, and mental.
However, traditional numerical interpretation methods have proven to be challenging to provide satisfying detection accuracies owing to the low quality of raw data especially under strong electromagnetic interference (EMI). To address this issue, in recent years, machine learning (ML) based techniques have been proposed to smartly map IMU captured data on disease detection and progress.
The authors:
(a) highlighted that after a decade of development, the combination of IMUs and ML algorithms for assistive disease diagnosis has become a hot topic, with an increasing number of studies reported yearly.
(b) to help readers to comprehensively understand the fundamentals and state-of-the-art techniques, proposed an article that systematically reviews this field, by introducing and explaining relevant application scenarios, discussing challenges, and predicting foreseeable future trends.
My opinion is that this article is interesting and very well written.
These are my minor comments:
1. Rearrange the abstract according to: background, methods, results, discussion and conclusion
2. In the present form, part of the review is in the introduction. I suggest rearranging the manuscript with the following structure:
a) A brief introduction.
b) A purpose with the answers to which the review aims to answer.
c) A short paragraph with the methods reporting databases and the strategy for the search.
d) All the found must go in the results
3. Insert a flow chart summarizing the themes of the results
4. Insert a table with the acronyms
Author Response
Point 1: Rearrange the abstract according to: background, methods, results, discussion and conclusion
Response 1: We have rearranged and added the methods, results, discussion, and conclusion part to the abstract.
Point 2: In the present form, part of the review is in the introduction. I suggest rearranging the manuscript with the following structure:
- a) A brief introduction.
- b) A purpose with the answers to which the review aims to answer.
- c) A short paragraph with the methods reporting databases and the strategy for the search.
- d) All the found must go in the results
Response 2: We have rearranged the review by:
- a) Emphasizing the dilemma of choosing a proper data processing model for IMU to tackle specific medical applications and providing three aspects that can clarify these problems in this review.
- b) Adding a section named “2. Methods and Taxonomy of Existing Approaches” explaining the article selection and the taxonomy of the selected articles.
- c) Summarizing the traditional and the mainstream ML methods for monitoring body movement based on the search results.
- d) Introducing detailed application scenarios with ML-assisted IMU monitoring from the view of different diseases.
Point 3: Insert a flow chart summarizing the themes of the results
Response 3: We have provided a summarization figure to show the popular model choice of different application scenario which covers the two major view of this review in figure 6.
Point 4: Insert a table with the acronymsam.
Response 4: We have provided a table with the acronyms in Appendix Section Table 6.